# Effects of Tertill® Weeding Robot on Weed Abundance and Diversity

**Kristine M. Averill [1], Anna S. Westbrook [1], Laura Pineda-Bermudez [2], Ryan P. O'Briant [1], Antonio DiTommaso [1] and Matthew R. Ryan [1,*]**

[1] Section of Soil and Crop Sciences, School of Integrative Plant Science, Cornell University, Ithaca, NY 14853, USA; averill.kristine@gmail.com (K.M.A.); asw265@cornell.edu (A.S.W.); rpo28@cornell.edu (R.P.O.); ad97@cornell.edu (A.D.)

[2] Section of Horticulture, School of Integrative Plant Science, Cornell University, Ithaca, NY 14853, USA; lp384@cornell.edu

* Correspondence: mrr232@cornell.edu

**Abstract:** Robotic weed control may reduce labor requirements, soil disturbance, and amount of herbicide applied relative to non-robotic methods. Tertill® is among the first weeding robots to become commercially available. This solar-powered robot moves in a random walk, avoiding obstacles using capacitive sensors, and cuts weeds with a string trimmer. We tested the effects of Tertill (two hours per week) with and without the string trimmer and hand weeding (from 3 to 5.6 min per week with a stirrup hoe) on weed communities at two field sites in Ithaca, NY. Tertill with trimmer and hand weeding provided similar levels of weed control (visual estimates averaging 2–9% ground cover at the end of the experiment, compared to 14–48% in the unweeded control). Without the string trimmer, Tertill was ineffective. Tertill did not significantly reduce monocot weed density but did reduce dicot weed density. At one site, Tertill reduced species richness and increased evenness based on density. Overall, these results suggest that Tertill can effectively remove newly emerged weed seedlings. Future research should investigate Tertill performance against more established weeds and the long-term effects of Tertill on weed community composition (e.g., possible selection for monocots and other species with low growing points).

**Keywords:** autonomous weed control; mechanical weed management; robotic weed control; weed community composition; weed community diversity





## 1. Introduction

Many weed control methods impose significant labor or capital costs, and some methods are also harmful to human or environmental health [1]. For example, hand weeding is time-intensive and physically demanding [2,3]. Intensive tillage requires a large amount of fuel and may threaten soil health [4]. Herbicides can be toxic to humans and other non-target organisms [5]. Other weed control methods, including physical, thermal, biological, and cultural methods, have drawbacks of their own.

Robotic weed control can help address challenges in weed management. Autonomous weeding robots often implement weeding methods already in use (e.g., spraying or mowing) without requiring direct human guidance [6]. This approach reduces labor requirements and the likelihood of injury or toxicity to humans. In addition, robotic weed control may be highly precise, reducing negative environmental impacts (e.g., spot-spraying of herbicides versus broadcast spraying [7]). Robotic weed control solutions are being developed for garden, horticultural, and agricultural contexts. Solutions developed in one context could be adapted for others. For example, a single robot might be used in a home garden, and a swarm of robots might be used in a field [8].

Tertill® (Tertill Corporation, North Billerica, MA, USA) is a small (approximately 12 cm tall by 21 cm diameter) solar-powered weeding robot intended for home gardens (up

to 18.6 m$^2$; Figure 1A). Tertill controls weeds through two mechanisms [9]. First, the robot travels in a random walk, using capacitive sensors to detect obstacles such as desirable plants. Desirable plants are distinguished from weeds according to a height threshold. Plants shorter than 2.5 cm are deemed weeds, and a spinning string trimmer is engaged, severing the weeds. Plants taller than this threshold are deemed desirable plants, and Tertill travels around them. Second, as Tertill travels, its wheels may disturb the soil and thereby reduce weed emergence and survival. If Tertill is introduced to a weed-free plot and allowed to run daily, these mechanisms are expected to prevent weeds from accumulating large root reserves [9] or adding to the soil seedbank. Weeds with "extensive or established roots" are difficult to kill with Tertill [9].

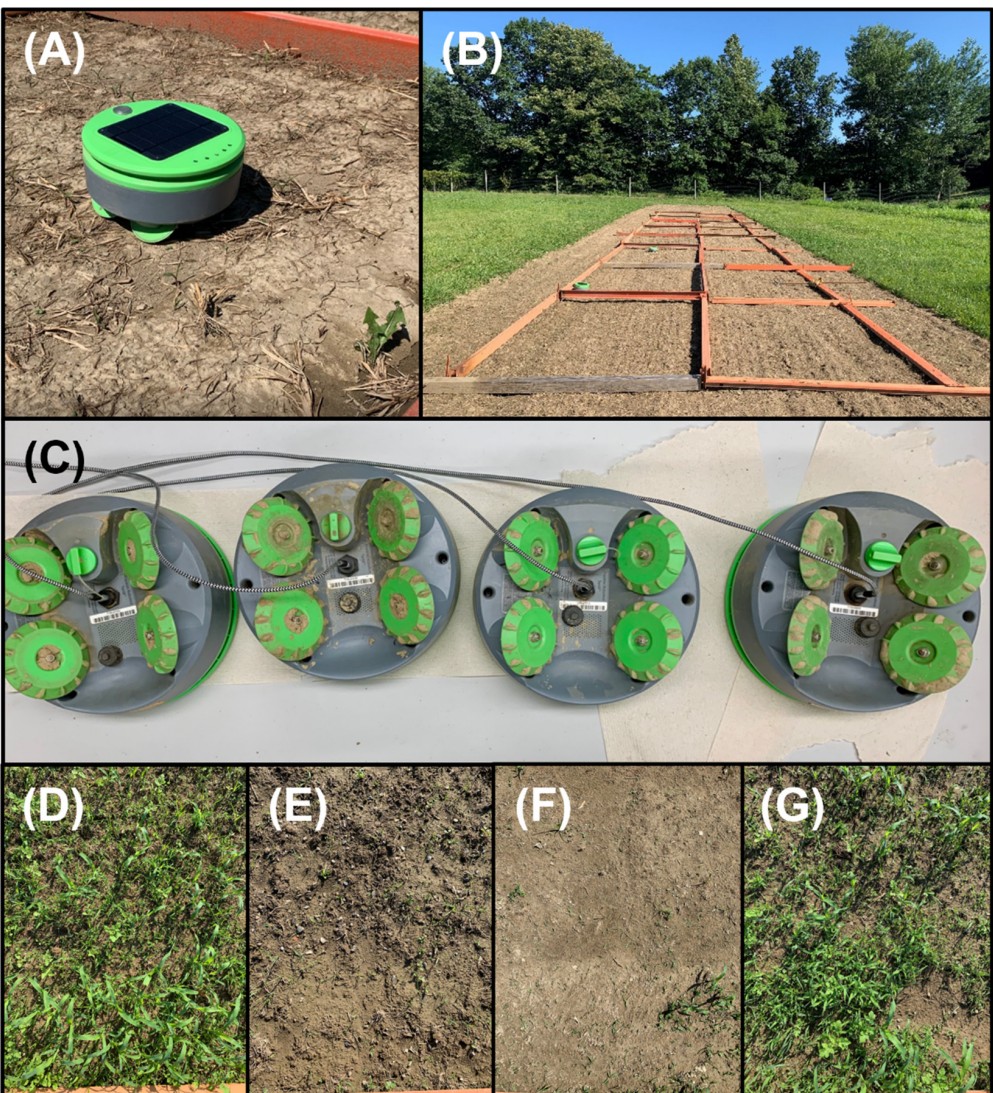

**Figure 1.** Tertill unit, experimental design, and plot photos. (**A**) Tertill travels in a random walk, using capacitive sensors to detect obstacles. Weed growth may be inhibited by its wheels and spinning string trimmer. (**B**) Old field site. Plots were 1.8 by 2.4 m. Four blocks of four treatments (T+, Tertill with trimmer; T−, Tertill without trimmer; HW, hand weeding by stirrup hoe; UWC, unweeded control) were set up at each location (old field and crop field) in Ithaca, NY. (**C**) Four Tertill units were used during the experiment. A pair of units [e.g., units 1 (T+) and 2 (T−) from left] was used in each field during each weeding session. Tertill units were charged from an electrical outlet using the USB port on their undersides. (**D**) Representative UWC plot at the old field site on August 20. (**E**) Representative HW plot at the old field site on August 20. (**F**) Representative T+ plot at the old field site on August 20. (**G**) Representative T− plot at the old field site on August 20.

Like any weed control method, Tertill may control some weeds more effectively than other weeds. Knowledge about this variation in weed control efficacy could facilitate the design of Integrated Weed Management (IWM) programs. For example, if Tertill is ineffective against particular species or functional groups, the uncontrolled species might become dominant and troublesome if land managers do not combine Tertill with other control methods that are effective against those species. With its spinning string trimmer, Tertill operates similarly to a mower. Autonomous mowers have been shown to provide effective, energy-efficient, and low-cost weed control, relative to conventional mowing [10]. Mowing may be relatively ineffective against weeds with resources stored belowground or low growing points, including some perennial or monocot weeds [4]. It is not yet clear whether Tertill favors such weeds, relative to annual broadleaf weeds, or whether this filtering affects weed community diversity.

Sanchez and Gallandt [11] conducted a greenhouse experiment to test whether Tertill reduced the densities of two surrogate weed species: condiment mustard [*Brassica juncea* (L.) Czern.] and pearl millet [*Pennisetum glaucum* (L.) R. Br.]. They also tested Tertill without the string trimmer to determine whether soil disturbance by the wheels contributed to weed control. When the string trimmer was removed, the efficacy of Tertill dropped from 60–72% to 4–39% for condiment mustard and from 54–75% to 16–29% for pearl millet. They concluded that the string trimmer (which severs weeds) and soil disturbance by the wheels (which may uproot or bury weeds) jointly contribute to monocot and dicot weed control. Although promising, these findings reflect the responses of only two species in a greenhouse setting.

A field experiment was conducted to test the effects of Tertill on weed cover, weed density, and weed community structure. It was hypothesized that Tertill without the trimmer would reduce weed cover and density, relative to an unweeded control, that Tertill with the trimmer would provide superior weed control, and that the efficacy of Tertill would vary with weed cotyledon type and life cycle, resulting in reduced species richness and evenness relative to the unweeded control.

## 2. Materials and Methods

### 2.1. Study Sites

The experiment was conducted over a three-week period from 30 July 2019 to 20 August 2019 at two locations in Ithaca, NY, USA: Cornell University's Crop Garden (42.451034 N, 76.459963 W) and Caldwell Field L (42.449561 N, 76.458312 W). Both fields had been used for crop production in previous years. However, the Crop Garden ("old field") had experienced greater crop and management diversity, including periods of no management, whereas Caldwell Field L ("crop field") had been tilled and planted more continuously. The old field was therefore expected to have a larger weed population and a more diverse weed community. Before the experiment began (first weeding on 30 July 2019), the two locations were rototilled and packed (Land Pride RTA2570, Land Pride, Salina, KS) to provide a smooth, even surface (29 July 2019). Four cover crop species were uniformly broadcast over the entire experimental area to supplement the existing weed soil seedbank (29 July 2019). These species were cereal rye (*Secale cereale* L.), red clover (*Trifolium pratense* L.), sorghum sudangrass (*Sorghum bicolor* (L.) Moench × *S. bicolor* var. *sudanense*), and white mustard (*Sinapis alba* L.). These "surrogate weeds" were seeded to ensure that each field contained enough newly emerged seedlings to accurately test the efficacy of Tertill. Because the cover crop species were small at the time of the experiment, they were treated as weeds by Tertill.

### 2.2. Experimental Design

At each location, the experiment was set up in a randomized complete block design with four blocks (Figure 1B). Treatment plots within blocks were 1.8 by 2.4 m with raised borders around all plots (old field) or Tertill treatment plots (crop field). Four treatment plots were included in each block: two Tertill treatments and two controls. The two Tertill treatments were distinguished by the presence (T+) or absence (T−) of the string trimmer.

In both these treatments, Tertill ran for one hour per plot twice per week (total of six hours over the duration of the experiment). To ensure that Tertill ran for the full hour, the Tertill developers provided customized software to allow the batteries to reach a lower shutoff point (49% cutoff rather than 69%). Otherwise, the settings on the production model were unchanged. Units were fully charged from an electrical outlet prior to the start of each weeding session. Ordinarily, Tertill is solar-powered and weeds for 1–2 h per day in intervals of 2–5 min [9]. Weeding sessions were conducted when the soil was dry to reduce the potential for mud interfering with wheel movement. Each weeding session was conducted with a single Tertill unit randomly chosen from a group of four units used for the experiment (Figure 1C). String trimmers were checked after each run and replaced once during the three-week experiment. The two control treatments were hand weeding with a stirrup hoe (HW) and an unweeded control (UWC). The stirrup hoe was used for a few minutes (from 3 to 5.6 min) per plot, once per week. The unweeded control did not receive any weed management.

### 2.3. Data Collection

On 20 August, the percentage ground cover of each weed species was visually estimated, and individuals of each weed species were counted within two 0.25 m$^2$ quadrats per treatment plot, then averaged across quadrats. Quadrats were placed randomly, except that they were not placed within 0.2 m of plot edges. Total weed cover was also measured using Canopeo, an application that measures fractional green canopy cover [12]. Canopeo measurements were taken by a single observer, who held a smartphone camera at shoulder height over the center of each plot. Canopeo measurements were taken twice: at the end of the experiment, i.e., three weeks after treatments were initiated (20 August), and approximately five weeks after the end of the experiment (26 September).

### 2.4. Data Analysis

Three seedling clumps could not be identified to species, so they were included in total cover and density measurements as "unknown dicot species" (two clumps) or "unknown monocot" (one clump). These unidentifiable clumps were excluded from analyses involving weed life cycle or weed community diversity. For analyses that focused on life cycle, the most common life cycle (annual, biennial, or perennial) for each species in our region was used. Only one species (wild carrot, *Daucus carota* L.) was identified as biennial, so we included biennial and perennial species in a single category.

Statistical tests were performed in R (version 4.1.0 [13]). Species richness was defined as the number of unique species identified in each plot. Evenness was defined as:

$$E_H = H/\ln(S) \tag{1}$$

where *H* is the Shannon Diversity Index and *S* is species richness. This index of evenness ranges from 0 to 1, with 0 representing complete dominance and 1 representing complete evenness. Visual estimates of weed cover by species (20 August) were used to calculate the Shannon Diversity Index (package "vegan").

Weed cover, density, richness, and evenness were evaluated using linear mixed models (packages "lme4", "lmerTest"). Location (old field or crop field), treatment (T+, T−, HW, or UWC), and their interaction were treated as fixed effects. Block was treated as a random effect. To meet model assumptions, some response variables were transformed: ln(y) for Canopeo cover on 20 August; logit(y) for Canopeo cover on September 26; $y^{0.5}$ for visual cover estimates, monocot density, dicot density, and species richness; $y^2$ for evenness based on cover; $y^3$ for evenness based on density. Pairwise differences between treatments were tested on the transformed response scales (packages "emmeans", "multcomp", $\alpha = 0.05$, Tukey method for *p* value adjustment). Graphs (package "ggplot2") show estimates and 95% confidence intervals back-transformed from the transformed scales.

## 3. Results

Location and weeding treatment both influenced weed cover. On August 20, three weeks after treatments were initiated, visual weed cover estimates were influenced by location ($p < 0.001$), weeding treatment ($p < 0.001$), and their interaction ($p = 0.002$). At the old field site, weed cover was lowest in the T+ (Tertill with trimmer) and HW (hand weeding) treatments (Figure 2A). Weed cover was higher in the T− (Tertill without trimmer) treatment and highest in the UWC (unweeded control) treatment. At the crop field site, weed cover was lower in T+ and HW compared to T− and UWC. Weed cover was generally higher at the old field site than at the crop field site. A similar pattern was detected in weed cover estimates from the Canopeo application (Figure 2B). Canopeo weed cover estimates were influenced by location ($p < 0.001$), weeding treatment ($p < 0.001$), and their interaction ($p < 0.001$). According to the Canopeo data, weed cover was lower in T+ and HW than T− and UWC at both sites. However, weed cover was lowest in HW at the old field site and lowest in T+ at the crop field site. On 26 September, more than one month after treatments concluded, weed cover in the T+ treatment was significantly lower than all other treatments at the crop field site (Figure 2C).

On August 20, weed density was affected by weeding treatment for monocot, dicot, annual, and biennial/perennial groups ($p < 0.01$; Figure 3). Monocot, annual, and biennial/perennial weed densities were affected by location ($p < 0.05$). Annual weed density was also affected by the interaction between location and weeding treatment ($p < 0.001$). At the old field site, monocot density was lower in HW than UWC; however, T+ and T− did not differ from HW or UWC (Figure 3A). Monocot weed density was very low and did not vary among treatments at the crop field site. In contrast, dicot density was lower in T+ and HW than UWC at both sites (Figure 3B). Annual weed density was lowest in T+ and HW, intermediate in T−, and highest in UWC at the old field site (Figure 3C). Annual weed density did not vary among treatments at the crop field site. Biennial/perennial weed density was lower in T+ than UWC at the old field site (Figure 3D). At the crop field site, biennial/perennial weed density was lower in T+ and HW than UWC.

A total of 44 species were identified across both locations. Weed species richness varied with location ($p < 0.001$) and weeding treatment ($p < 0.001$; Figure 4). Pairwise comparisons did not reveal significant differences among weeding treatments at the old field site. At the crop field site, the T+ treatment had lower richness than the T− and UWC treatments; T+ and HW did not differ (Figure 4A). An evenness index based on weed cover varied with location ($p = 0.01$) but did not vary with weeding treatment ($p = 0.1$; Figure 4B). However, the interaction between location and weeding treatment influenced a similar evenness index based on weed density ($p = 0.006$; Figure 4C). At the crop field site, density-based evenness was higher in T+ compared to the other treatments.

The old field had different dominant species than the crop field (Table 1). In the old field, only one of the four species seeded for this experiment (sorghum sudangrass) was among the five most dominant species. In the crop field, all four species were represented. The three most dominant species in the old field were monocots, whereas the three most dominant species in the crop field were dicots.

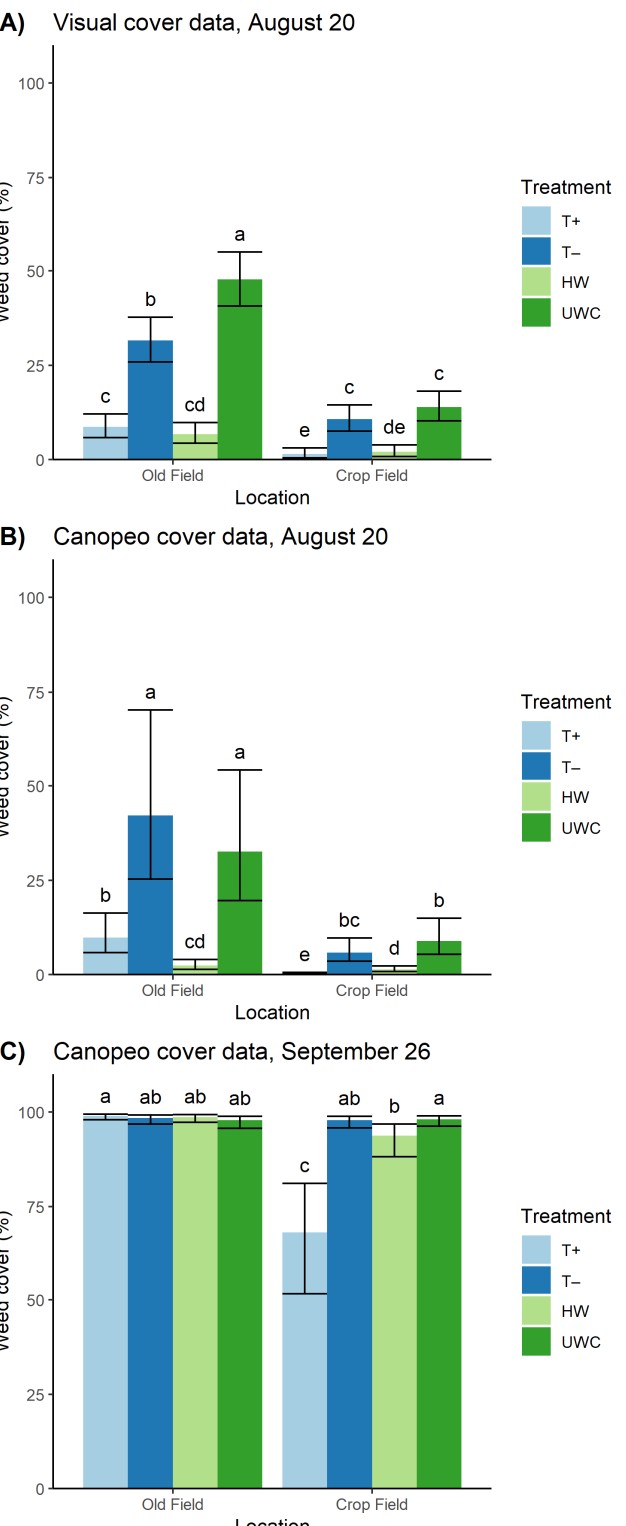

**Figure 2.** Effects of weeding treatment and location on weed cover. (**A**,**B**) Weed cover estimated visually or by Canopeo on 20 August 2019. (**C**) Weed cover estimated by Canopeo on 26 September 2019. Bars marked with the same letter are not significantly different according to a Tukey test ($\alpha$ = 0.05); 95% confidence intervals are shown. Abbreviations: T+—Tertill with trimmer; T−—Tertill without trimmer; HW—hand weeding by stirrup hoe; UWC—unweeded control.

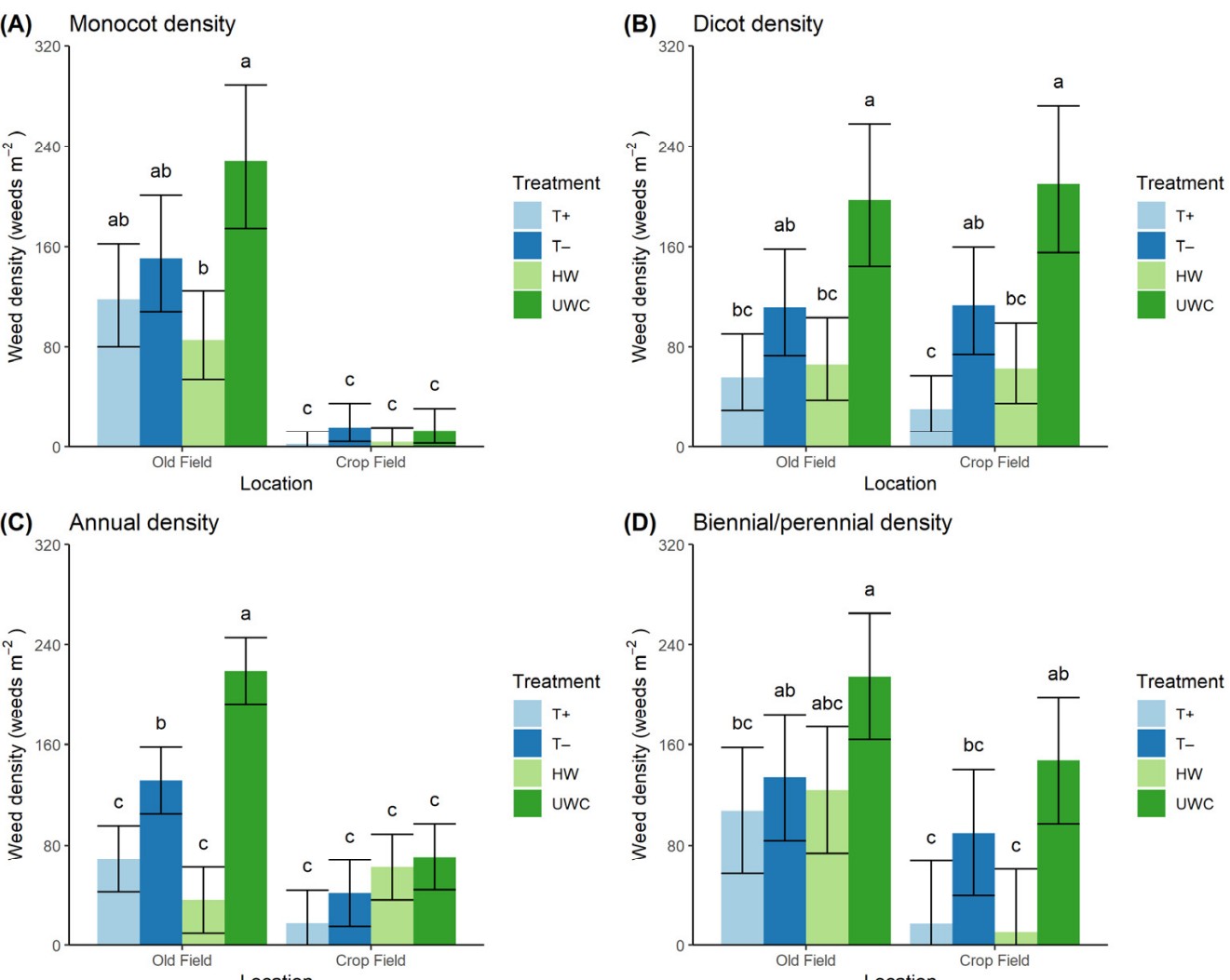

**Figure 3.** Effects of weeding treatment and location on weed density by (**A**,**B**) cotyledon type and (**C**,**D**) life cycle. Weed density was estimated by species on 20 August 2019; then these species-specific densities were aggregated by cotyledon type or life cycle. Bars marked with the same letter are not significantly different according to a Tukey test ($\alpha = 0.05$); 95% confidence intervals are shown. Abbreviations: T+—Tertill with trimmer; T−—Tertill without trimmer; HW—hand weeding by stirrup hoe; UWC—unweeded control.

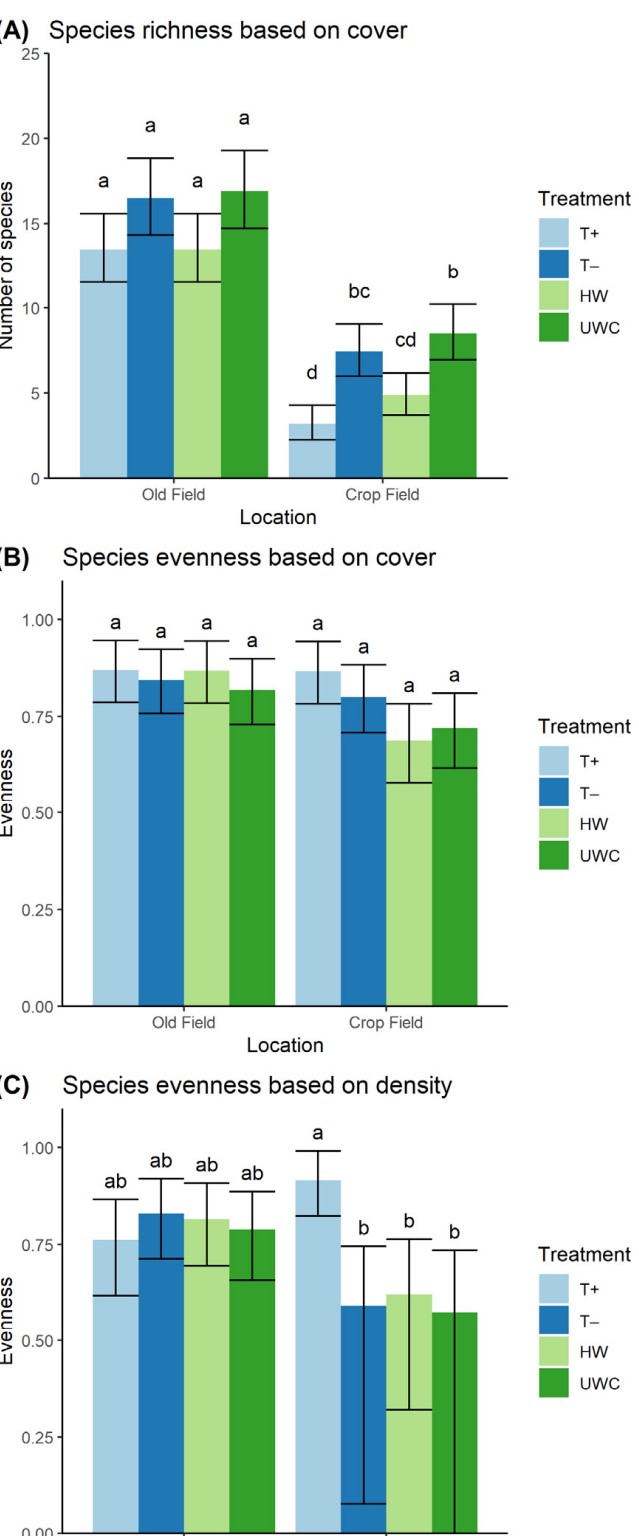

**Figure 4.** Effects of weeding treatment and location on species richness and evenness. (**A**) Total species richness per plot. (**B**,**C**) Evenness reflects whether weed species contribute equally to total weed cover (visual estimate) or density. This index ranges from 0 (complete dominance) to 1 (complete evenness). Bars marked with the same letter are not significantly different according to a Tukey test ($\alpha = 0.05$); 95% confidence intervals are shown. Abbreviations: T+—Tertill with trimmer; T−—Tertill without trimmer; HW—hand weeding by stirrup hoe; UWC—unweeded control.

**Table 1.** Dominant weed species in the old field and crop field in Ithaca, NY. Within each location, the five species shown had the highest percent cover (sum across plots). Percent cover was estimated visually on 20 August 2019. Abbreviations: T+—Tertill with trimmer; T−—Tertill without trimmer; HW—hand weeding by stirrup hoe; UWC—unweeded control.

| | | | | Percent Cover (Median, Mean ± SE) | | | |
|---|---|---|---|---|---|---|---|
| **Old Field** | | | | | | | |
| **Common Name** | **Scientific Name** | **Cotyledon Type** | **Life Cycle** | **T+** | **T−** | **HW** | **UWC** |
| Smooth crabgrass | *Digitaria ischaemum* | Monocot | Annual | 2.0, 2.0 ± 0.8 | 3.3, 3.4 ± 1.1 | 0.4, 0.4 ± 0.2 | 6.8, 6.5 ± 1.3 |
| Yellow nutsedge | *Cyperus esculentus* | Monocot | Perennial | 1.1, 1.1 ± 0.2 | 5.0, 4.3 ± 1.1 | 0.9, 1.0 ± 0.3 | 5.3, 5.7 ± 1.0 |
| Sorghum sudangrass * | *Sorghum bicolor × S. bicolor* var. *sudanense* | Monocot | Annual | 1.0, 0.8 ± 0.3 | 3.0, 4.0 ± 1.2 | 0.0, 0.0 ± 0.0 | 7.3, 7.1 ± 3.1 |
| Redroot pigweed | *Amaranthus retroflexus* | Dicot | Annual | 0.0, 0.3 ± 0.3 | 3.5, 3.3 ± 1.4 | 0.0, 0.2 ± 0.2 | 2.8, 4.9 ± 3.0 |
| Large crabgrass | *Digitaria sanguinalis* | Monocot | Annual | 0.4, 0.5 ± 0.2 | 5.0, 5.0 ± 1.7 | 0.2, 0.1 ± 0.1 | 1.6, 2.6 ± 1.3 |
| **Crop Field** | | | | | | | |
| **Common Name** | **Scientific Name** | **Cotyledon Type** | **Life Cycle** | **T+** | **T−** | **HW** | **UWC** |
| Red clover * | *Trifolium pratense* | Dicot | Perennial | 0.4, 1.0 ± 0.7 | 4.0, 3.9 ± 0.1 | 0.1, 0.1 ± 0.1 | 5.5, 5.5 ± 1.2 |
| Wormseed mustard | *Erysimum cheiranthoides* | Dicot | Annual | 0.3, 0.4 ± 0.1 | 2.0, 1.9 ± 0.3 | 1.0, 1.5 ± 0.7 | 1.8, 1.6 ± 0.4 |
| White mustard * | *Sinapis alba* | Dicot | Annual | 0.0, 0.1 ± 0.1 | 1.3, 1.1 ± 0.2 | 0.0, 0.0 ± 0.0 | 3.8, 3.6 ± 1.5 |
| Sorghum sudangrass * | *Sorghum bicolor × S. bicolor* var. *sudanense* | Monocot | Annual | 0.0, 0.1 ± 0.1 | 2.5, 2.5 ± 0.2 | 0.0, 0.0 ± 0.0 | 0.6, 0.7 ± 0.3 |
| Cereal rye * | *Secale cereale* | Monocot | Annual | 0.0, 0.0 ± 0.0 | 0.3, 0.4 ± 0.2 | 0.0, 0.1 ± 0.1 | 0.6, 0.8 ± 0.2 |

* Species sown as surrogate weeds on 29 July 2019.

## 4. Discussion

Overall, the results support our first hypothesis that Tertill would reduce weed cover and density, especially with the trimmer. At the conclusion of the experiment, weed cover in the T+ treatment was reliably lower than weed cover in the UWC treatment. The T+ treatment also reduced the densities of dicot weeds (both sites), annual weeds (old field), and biennial/perennial weeds (both sites) compared to the UWC treatment. However, weed cover and density were typically indistinguishable between T− and UWC (except for visual cover and annual weed density at the old field site). This finding suggests that the effects of Tertill on weeds are almost entirely attributable to the string trimmer. In contrast, Sanchez and Gallandt [11] reported that the Tertill wheels contributed 16% efficacy in the condiment mustard trial and 22% efficacy in the pearl millet trial. Their single-species trials were conducted in an indoor experimental arena on a layer of field soil prepared with a bed-shaping rake. It seems likely that the soil in these trials was looser and therefore more easily disturbed by Tertill wheels than the field soils in our outdoor experiment.

In general, the HW and T+ treatments provided similar levels of weed control (Figure 1D–F), suggesting that Tertill may be a viable alternative to hand weeding. Hand weeding is a common, time-intensive, and physically demanding task in garden and agri-

cultural contexts [2,3,14]. Tertill could make gardening more accessible for people with limited time or mobility, although its current price (USD 349 per unit [9]) may limit availability to prospective gardeners. A modified version of Tertill could be developed for agricultural contexts [11]. An agricultural Tertill unit should be able to maneuver in tighter spaces, such as the spaces within crop rows. In addition, some agricultural applications would require Tertill to navigate uneven, rocky, or muddy ground. Agricultural soils can also be dry or hard, potentially eliminating any weed-control benefit of the Tertill wheels. Lastly, individual Tertill units should be able to communicate with other Tertill units to effectively cover the entire field area.

We found some support for our second hypothesis that Tertill would select for particular weed species traits (Figure 3). Tertill did not reduce monocot density at either site. In contrast, Tertill with the string trimmer did reduce dicot density as effectively as hand weeding at both sites. This result contrasted with the results of Sanchez and Gallandt [11], who reported that Tertill had similar efficacy against condiment mustard and pearl millet. Possible reasons for this discrepancy include the greater weed community diversity in our experiment and the outdoor setting with different soil conditions. In other studies, monocot weeds have proven difficult to kill by aboveground cutting. For example, McCool et al. [15] reported that a cutting tool (whipper-snapper) did not effectively control monocots, but monocots were largely controlled by an autonomous weeding robot fitted with a tine or arrow hoe. Brown and Gallandt [16] have demonstrated that weed control may be improved through the 'stacking' of multiple mechanical tools. Combining Tertill with other mechanical tools or non-mechanical approaches would likely improve monocot control.

Both annual and biennial/perennial weeds were affected by Tertill (Figure 3C,D). At the old field site, Tertill reduced annual weed density even when operated without the string trimmer. The small annual weed community at the crop field site did not respond to any weeding treatment. Biennial/perennial weeds were reduced by Tertill (only with the string trimmer) at both sites. Thus, our results do not clearly indicate whether Tertill favors particular weed life cycles.

Cover and density estimates revealed that weed abundance was higher at the old field site than at the crop field site (Figures 2 and 3). Species richness was also higher at the old field site (Figure 4). These differences are consistent with a history of greater management diversity at the old field site, including periods of no management. Tertill did not affect species richness or evenness at the old field site (Figure 4). At the crop field site, the T+ and HW treatments both reduced species richness (in agreement with our second hypothesis). The T+ treatment increased density-based evenness at the crop field site (contrary to our second hypothesis). This finding might reflect selectivity for species that occurred at high density in the UWC treatment at the crop field site, which were presumably dicots (Figure 3A). Overall, however, these results do not provide strong evidence that short-term Tertill weeding alters weed community structure.

Our study did not address long-term effects of Tertill on weed community structure. Although we measured total weed cover approximately five weeks after the experiment ended (Figure 2C), we did not test whether weed species differed in their ability to regrow after the experiment. We would expect that more monocot and perennial weeds survived Tertill weed management due to their low growing points and belowground reserves. In the long term, Tertill might shift weed community composition in favor of these groups. This hypothesis requires testing in multiyear studies. Conventional mowing programs are known to select for particular weed traits [4]; however, Tertill weeding differs from most of these mowing programs in that weeds are trimmed much more frequently. Although Tertill weeding occurred twice per week in our experiment, Tertill is intended to run daily [9]. Frequent trimming by Tertill may deplete ground-level and belowground reserves over time. Therefore, the long-term effects of Tertill on weed community structure remain largely unknown and cannot be easily inferred from the effects of other mechanical weed control methods.

## 5. Conclusions

Our findings confirm that Tertill provides effective control of newly emerged weed seedlings. Using a field experiment at two sites, we have shown that Tertill reduces weed cover and density. Given these results, future research might focus on performance against more established weeds. Performance at larger field scales would be another useful direction for future research, especially if this research could be conducted with a modified Tertill unit designed specifically for agricultural contexts. However, we emphasize that no single tactic constitutes a complete weed management program. Tertill is expected to shift weed communities in favor of species that are more resistant to this tool, such as monocot weeds. In garden settings, weeding robots might be combined with tactics such as occasional hoeing to remove belowground organs. In agricultural settings, growers who use weeding robots should continue to consider management diversity and ecological principles as well as precision and labor efficiency.

**Author Contributions:** Conceptualization, K.M.A., A.D. and M.R.R.; methodology, K.M.A., A.D. and M.R.R.; formal analysis, A.S.W. and K.M.A.; resources, A.D. and M.R.R.; writing—original draft preparation, A.S.W., L.P.-B. and R.P.O.; writing—review and editing, all authors; funding acquisition, A.D. and M.R.R. All authors have read and agreed to the published version of the manuscript.

**Funding:** This research was supported by the joint research and extension project 2020-21-210 funded by the Cornell University Agricultural Experiment Station (Hatch funds) and Cornell Cooperative Extension (Smith Lever funds) received from the National Institutes of Food and Agriculture (NIFA) U.S. Department of Agriculture (USDA). Any opinions, findings, conclusions, or recommendations expressed in this publication are those of the authors and do not necessarily reflect the view of USDA.

**Institutional Review Board Statement:** Not applicable.

**Informed Consent Statement:** Not applicable.

**Data Availability Statement:** Data are available upon request.

**Acknowledgments:** We thank Joe Jones and Jeffrey Vandegrift of Franklin Robotics. We thank Scott Morris and María Paula Osuna Barreto for field and robot operation assistance. We thank Gene Sczepanski and Kathy Howard for help securing field locations and Cornell Farm Services for field preparation.

**Conflicts of Interest:** The authors declare no conflict of interest.

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
