# Peer review of "Effects of Tertill® Weeding Robot on Weed Abundance and Diversity"

_agronomy, doi:10.3390/agronomy12081754_

Round 1
Reviewer 1 Report
Effects of Tertill® weeding robot on weed abundance and diversity
Line 21 ‘amount of herbicide applied’ not ‘herbicide application rates’. A legislated recommended rate is a legal requirement
Line 38-44 This paragraph needs rewriting as there is no flow, it reads like a group of dot points
Line 54 What are the dimensions of the Tertill® unit? Needs to be stated somewhere, probably here at first mention
Figure 1 Images of the four units (Figure 1C) are unnecessary
Line 79-81 The uncontrolled species may become the dominant species leading to other issues
Line 91-93 Delete this sentence, the details are repeated in the next two sentences.
Line 94 Change ‘dropped to 4% to 39%’ to ‘dropped to between 4% and 39%’
Line 94 Replace ‘The authors’ with ‘They’
Line 99 Write in third person. Change ‘We conducted a field experiment to test…’ to ‘A field experiment was conducted to…’
Line 100 Change ‘We’ to ‘It was’
Line 102 Change ‘control. We also hypothesised that…’ to ‘control and that…’
Line 115 What equipment was used to rototill and smooth the area?
Line 117 How were these species seeded? In rows using a drill or spread and raked in? Were they expected to be controlled ie. treated as weeds or were they treated as desirable species
Line 122 Change ‘according to a’ to ‘in a’
Line 135 Delete ‘(Figure 1C)’
Line 142 Change ‘cover’ to ‘ground cover %’
Line 144 How far from the edge?
Line 147 Add ‘twice;’ after ‘taken’
Line 185 Any statistics for this statement?
Line 190-2 Maybe reword this sentence
Line 224 Move ‘(Figure 4A)’ from Line 223 to here
Line 231-2 Delete sentence define richness, already stated Line 158-9
Line 238-41 First three sentences belong in the discussion
Line 241 Delete ‘also’
Table 1 Define which were the sown species with a symbol
Line 264 Was it also smoother?
Some unanswered questions that should be discussed
· Is the Tertill® better suited for in-crop weed control or fallow situations, can it move between rows, with no dimensions provided we don’t know
· How much area can it cover in a given time or battery life, this determines it suitability for commercial areas
Author Response
We appreciate the thoughtful comments provided and have prepared a point-by-point response below. Our responses are in bold red font.
Line 21: ‘amount of herbicide applied’ not ‘herbicide application rates’. A legislated recommended rate is a legal requirement
We have made this change in the revised manuscript.
Line 38-44: This paragraph needs rewriting as there is no flow, it reads like a group of dot points
We have revised the first sentence to clarify the structure of this introductory paragraph (a generalization followed by several examples).
Line 54: What are the dimensions of the Tertill® unit? Needs to be stated somewhere, probably here at first mention
The dimensions are now provided at first mention.
Figure 1: Images of the four units (Figure 1C) are unnecessary
We prefer to include panel 1C because it shows the two possible mechanisms of weed control (string trimmer and wheels) and how units were charged using the USB charging port on their underside. The caption was revised for clarity.
“C) Four Tertill units were used during the experiment. A pair of units [e.g., units 1 (T+) and 2 (T-) from left] was used in each field during each weeding session. Tertill units were charged from an electrical outlet using the USB port on their underside.”
Line 79-81: The uncontrolled species may become the dominant species leading to other issues
We have rephrased this sentence. We wish to express the idea that weed management tactics do filter weed communities, but growers (or gardeners) who are prepared for shifts in weed community composition can apply integrated approaches to minimize yield loss.
Line 91-93: Delete this sentence, the details are repeated in the next two sentences.
We have revised this paragraph so that all values are presented in a single sentence.
Line 94: Change ‘dropped to 4% to 39%’ to ‘dropped to between 4% and 39%’
For clarity, the revised version uses en dashes.
Line 94: Replace ‘The authors’ with ‘They’
We have made this change.
Line 99: Write in third person. Change ‘We conducted a field experiment to test…’ to ‘A field experiment was conducted to…’
We have made this change.
Line 100: Change ‘We’ to ‘It was’
We have made this change.
Line 102: Change ‘control. We also hypothesised that…’ to ‘control and that…’
We have made this change.
Line 115: What equipment was used to rototill and smooth the area?
Information about the equipment was added to the revised manuscript.
Line 117: How were these species seeded? In rows using a drill or spread and raked in? Were they expected to be controlled ie. treated as weeds or were they treated as desirable species
We have clarified that these species were broadcast and treated as weeds.
Line 122: Change ‘according to a’ to ‘in a’
We have made this change.
Line 135: Delete ‘(Figure 1C)’
Please see comment on Figure 1 above.
Line 142: Change ‘cover’ to ‘ground cover %’
We use “percentage ground cover”.
Line 144: How far from the edge?
Requested information was added to the revised manuscript.
Line 147: Add ‘twice;’ after ‘taken’
We have made this change.
Line 185: Any statistics for this statement?
We have revised this sentence to more clearly express the idea that treatment effects on ground cover were similar regardless of which method was used to estimate cover (188–189). Statistics supporting this conclusion are given in the following three sentences.
Line 190-2: Maybe reword this sentence
We have made this change.
Line 224: Move ‘(Figure 4A)’ from Line 223 to here
We have made this change.
Line 231-2: Delete sentence define richness, already stated Line 158-9
We have made this change.
Line 238-41: First three sentences belong in the discussion
We have moved these sentences to the Discussion.
Line 241: Delete ‘also’
We have made this change.
Table 1: Define which were the sown species with a symbol
We have made this change.
Line 264: Was it also smoother?
It is plausible that soil was smoother in the indoor experiment conducted by Sanchez and Gallandt, relative to our experiment; however, we do not have any way to verify this assumption. We focus on the looseness/compaction of the soil because this property, more than surface smoothness, would determine whether Tertill wheels disturbed the soil and contributed to weed suppression.
Some unanswered questions that should be discussed
- Is the Tertill® better suited for in-crop weed control or fallow situations, can it move between rows, with no dimensions provided we don’t know
The revised manuscript provides dimensions and notes that Tertill may currently be unable to move between rows, which would limit its utility in agricultural contexts.
- How much area can it cover in a given time or battery life, this determines it suitability for commercial areas
We have clarified that solar power usually enables Tertill to weed for 1–2 hours per day. This amount of weeding is sufficient to maintain a home garden up to 18.6 m2. Coordination among multiple units would likely be required for agricultural use of this technology.
Reviewer 2 Report
The paper entitled “Effects of Tertill weeding robot on weed abundance and diversity” has been reviewed. The paper deals on testing a weeding machine that is being commercialised for home gardening in a much larger scale, that is, for agricultural fields, to check its efficacy to kill weeds as an alternative to the chemical weed control methods.
The paper is very well written, structured and easy to follow. The introduction has general statements about the field of research in order to provide the reader a context for the problem, it also has more specific statements concerning the machine that will be tested giving data of previous trials and gives the objectives of the study and the hypothesis involved. The methodology is well described and the results are clearly shown with sound analysis.
I would recommend to the authors don’t box figures and remove horizontal and vertical lines in the figures as they are mainly used in a communication to provide the reader with the overall trend of the data or show the magnitude of the response but not the exact numbers (for this you can display a table). Additionally, legend of figure 3 lacks to explain when weed density was estimated.
This is a very short experiment, only 3-week duration, so it cannot give further insights on the effect of the machine on the weed community structure. Consequently, and taking into account the intended use of this machine for weeding large commercial fields, I would recommend to further discuss the potential limitations of the machine for this agricultural use. For example, what is the efficacy expected in crops that are seeded in rows (i.e. cereals) where the machine could not easily change between rows easily? What can be the effect of the characteristics of the soil on the efficacy? I guess that the machine needs to be in a flat-non-stony field for a proper functioning … which is not always possible. An additional fact mentioned by the authors when comparing its results with other researcher but not further mentioned in the discussion is that the effect of the machine on the weeds seems to be due mainly to the string trimmer, which means that the soil must have a certain degree of moisture (softness) to be altered. However, many agricultural fields are located in regions with drought periods, irregular rainfall pattern than do not favour to have a “soft” soil; furthermore, direct drill sowing is increasingly being implemented, which favours hard soils. How these facts can affect the efficacy of the machine?
Author Response
Reviewer 1 Comments
Please see our responses in red below.
The paper entitled “Effects of Tertill weeding robot on weed abundance and diversity” has been reviewed. The paper deals on testing a weeding machine that is being commercialised for home gardening in a much larger scale, that is, for agricultural fields, to check its efficacy to kill weeds as an alternative to the chemical weed control methods.
The paper is very well written, structured and easy to follow. The introduction has general statements about the field of research in order to provide the reader a context for the problem, it also has more specific statements concerning the machine that will be tested giving data of previous trials and gives the objectives of the study and the hypothesis involved. The methodology is well described and the results are clearly shown with sound analysis.
I would recommend to the authors don’t box figures and remove horizontal and vertical lines in the figures as they are mainly used in a communication to provide the reader with the overall trend of the data or show the magnitude of the response but not the exact numbers (for this you can display a table). Additionally, legend of figure 3 lacks to explain when weed density was estimated.
We have removed the boxes and gridlines from Figures 2–4 and added the sampling date (August 20) to the Figure 3 legend (217).
This is a very short experiment, only 3-week duration, so it cannot give further insights on the effect of the machine on the weed community structure. Consequently, and taking into account the intended use of this machine for weeding large commercial fields, I would recommend to further discuss the potential limitations of the machine for this agricultural use. For example, what is the efficacy expected in crops that are seeded in rows (i.e. cereals) where the machine could not easily change between rows easily? What can be the effect of the characteristics of the soil on the efficacy? I guess that the machine needs to be in a flat-non-stony field for a proper functioning … which is not always possible. An additional fact mentioned by the authors when comparing its results with other researcher but not further mentioned in the discussion is that the effect of the machine on the weeds seems to be due mainly to the string trimmer, which means that the soil must have a certain degree of moisture (softness) to be altered. However, many agricultural fields are located in regions with drought periods, irregular rainfall pattern than do not favour to have a “soft” soil; furthermore, direct drill sowing is increasingly being implemented, which favours hard soils. How these facts can affect the efficacy of the machine?
We have expanded the section of the Discussion on agricultural applications (273–279). The revised version mentions several possible issues, including the potential for reduced efficacy within crop rows or under dry conditions. We also added a sentence to the Conclusions to clarify that the Tertill model we tested (designed for home gardens) is not intended or ready for agricultural use (326–328).
Reviewer 3 Report
Authors tested Tertill, a commercially available robotic solar-powered weeder intended for use in home gardens. The robot travels in a random walk, and plants shorter than 2.5 cm are deemed weeds and a spinning string trimmer is engaged, severing the weeds.
Tertill has variable efficacy on annual monocot and dicot weeds. Though, it can be efficiently used for weed control in home gardens. The disadvantage of this weeder is that established weeds longer than 3.0 cm will not be controlled (line 64). Tertill with the trimmer provided superior weed control, and it may be a viable alternative to hand weeding.
Authors have highlighted future research areas in lines 302-325. I agree with them.
Lines 116-120: Four cover crop species were uniformly seeded over the entire experimental area to supplement the existing weed soil seedbank (July 29, 2019). What is the purpose of cover crop? Kindly clarify how these supplemented the weed soil seedbank. Was these cover crops considered as weeds by Tertill?
Author Response
Reviewer 2 Comments
Please see our response in red below.
Authors tested Tertill, a commercially available robotic solar-powered weeder intended for use in home gardens. The robot travels in a random walk, and plants shorter than 2.5 cm are deemed weeds and a spinning string trimmer is engaged, severing the weeds.
Tertill has variable efficacy on annual monocot and dicot weeds. Though, it can be efficiently used for weed control in home gardens. The disadvantage of this weeder is that established weeds longer than 3.0 cm will not be controlled (line 64). Tertill with the trimmer provided superior weed control, and it may be a viable alternative to hand weeding.
Authors have highlighted future research areas in lines 302-325. I agree with them.
Lines 116-120: Four cover crop species were uniformly seeded over the entire experimental area to supplement the existing weed soil seedbank (July 29, 2019). What is the purpose of cover crop? Kindly clarify how these supplemented the weed soil seedbank. Was these cover crops considered as weeds by Tertill?
We have added two sentences to clarify that the cover crops were treated as weeds by Tertill (120–122). The cover crops were used to increase the number of seedlings emerging at these recently tilled sites. Testing the effects of Tertill on a very sparse plant community might have yielded inconclusive or unreliable results.